# Diversity-Generating Retroelements in Prokaryotic Immunity

**DOI:** 10.3390/ijms24065614

**Published:** 2023-03-15

**Authors:** Ilya S. Belalov, Arseniy A. Sokolov, Andrey V. Letarov

**Affiliations:** 1Laboratory of Microbial Viruses, Winogradsky Institute of Microbiology RC Biotechnology RAS, 117312 Moscow, Russia; ars.a.sokolov@gmail.com; 2Department of Biological and Medical Physics, Moscow Institute of Physics and Technology, Institutskiy per. 9, 141701 Moscow, Russia

**Keywords:** diversity-generating retroelements, adaptive immunity, prokaryotic immunity, somatic hypermutation, metagenomics

## Abstract

Adaptive immunity systems found in different organisms fall into two major types. Prokaryotes possess CRISPR-Cas systems that recognize former invaders using memorized (captured) pieces of their DNA as pathogen signatures. Mammals possess a vast repertoire of antibodies and T-cell receptor variants generated in advance. In this second type of adaptive immunity, a pathogen presentation to the immune system specifically activates the cells that express matching antibodies or receptors. These cells proliferate to fight the infection and form the immune memory. The principle of preemptive production of diverse defense proteins for future use can hypothetically take place in microbes too. We propose a hypothesis that prokaryotes employ diversity-generating retroelements to prepare defense proteins against yet-unknown invaders. In this study, we test this hypothesis with the methods of bioinformatics and identify several candidate defense systems based on diversity-generating retroelements.

## 1. Introduction

Bacteria have developed numerous defense mechanisms, many of which use distinct chemical signals to identify an intruder and activate a response that either inhibits phage infection or induces cell death to safeguard the microbial population from pathogen spread [1]. The only known type of adaptive immunity in prokaryotes is the CRISPR-Cas system [2]. The basic principle of the CRISPR-Cas system is memorizing invaders using fragments of their DNA, captured as so-called spacers into CRISPR arrays. The RNA transcripts of these arrays, in the form of guide RNA, target the interfering proteins to the DNA or RNA of invaders carrying the sequences of the spacers. Vertebrate immune systems, on the other hand, generate a gigantic pool of immunity sensors (antibodies and T-cells receptors) before contact with any pathogen [3]. Pathogen invasion induces the proliferation of cells capable of producing antibodies or receptors that fit the pathogen antigens. These activated immune cells fight the pathogen and form the immune memory, facilitating a more rapid response if the pathogen returns. Among the means of fine-tuning the immune sensors in the activated lymphocyte clones, vertebrates have developed the mechanism of somatic hypermutation. Hypermutation, mediated by specific mechanisms, is known in prokaryotes as well [4,5]. However, there is no known defense system employing an analog of somatic hypermutation in bacteria and archaea.

Recent research suggests that various molecular mechanisms, formerly believed to exist solely within eukaryotic organisms, actually evolved from bacterial and archaeal ancestors. These findings challenge prior assumptions and suggest a more complex evolutionary history than previously understood, with significant implications for our comprehension of the origins and development of complex cellular processes [6]. However, there is no known defense system employing analogs of clonal selection or somatic hypermutation in bacteria and archaea. If such a system exists, it must include a diversity-generating mechanism that would ensure a large repertory of the target-recognition elements prior to the invader’s attack (analogous to B and T cells’ clonal variability in an immunologically naïve animal organism). Additionally, it may (optionally) further modify the selected target-recognition elements in the clones selected by the invader pressure to generate the best defense against this particular invader. We hypothesize about adaptive prokaryotic immunity systems, which employ diversity-generating retroelements (DGRs) in order to produce diverse antiviral proteins.

It is worth noting that, unlike many laboratory settings in which stringent phage selection is applied (for example, as in the classical Luria and Delbruck fluctuation test experiment [7]), in the natural environment, mild selective pressure of phage infection is more frequent. Under such conditions, different pathways of resistance development may be facilitated, including the selection of partially resistant clones with a subsequent strengthening of the resistance in the course of local host–pathogen coevolution [8,9]. This mechanism appears to be more compatible with the suggested mode of action of a hypothetical system of prokaryotic defense based on clonal selection and somatic hypermutation.

In our search for the known molecular mechanism that may satisfy the above-specified requirements, we hypothesized that adaptive prokaryotic immunity systems may employ diversity-generating retroelements (DGRs) to produce diverse antiviral proteins. DGR systems include four essential components: reverse transcriptase (RT), template repeat (TR), variable repeat (VR), and accessory variability determinant protein (Avd) [10,11]. The sequences of TR and VR are very similar to each other. Hypermutation occurs during RT-mediated cDNA synthesis from transcribed TR. Mutated TR copy replaces homologous VR in a process referred to as retrohoming. The Avd protein is essential to retrohoming and is encoded by the majority of known DGR systems [11,12]. Such events occur independently in different cells creating a large diversity of VR variants at the population level. The utility of the VR alterations depends on the function of the VR-containing protein [10,13,14]. In the *Bordetella* phage BPP-1, the archetypal DGR system [15], the VR is located within the major tropism determinant (*mtd*) gene, encoding the tail fiber protein. Multiple mutations in this protein enable adaptation to the host’s phase variations of the surface molecules via the viral tropism switch. Our hypothesis suggests that it may be possible to find hypervariable genes in prokaryotes constructed from a VR with functions in antiviral immunity. This defense system would then play a role in producing preadaptations to new (or mutated) viruses.

The defense machinery employed by microorganisms includes a vast variety of proteins and protein systems [16,17], the majority of which are likely yet to be described. Different types of defense systems are often co-localized in genomes, forming so-called ‘immunity islands’ [18]. This association has created a basis for the successful identification of new systems of prokaryotic defense [19,20]. Using an analogous approach, in this study, we collect evidence in support of the hypothesis that prokaryotes organize defense mechanisms around DGR systems.

## 2. Results

We aimed for the comprehensive identification of new prokaryotic defense systems, utilizing DGRs, in the microbial genomes. For each of the 32,321 publicly available DGR-encoding nucleotide sequences [12,13,21,22], we explored the vicinity of two DGR components—VR and RT—for the genes encoding immunity-related protein families (Pfams). To accomplish this for each Pfam occurring within 10 kbp from the above-mentioned DGR components (1,348,709 proteins and 4907 Pfams in total), we computed the Icity score [23], which is a metric reflecting the probability of a functional association between the given Pfam and the components of DGR systems based on the co-localization statistics (see Section 4 and [23]). Due to intrinsic stochasticity embedded in the Icity protocol, results of different runs do not have to coincide exactly. For this reason, we averaged seven runs for each Pfam (see Section 4). High Icity score values occur only for a fraction of Pfams (see the histogram in Figure 1), 435 of which had an average Icity score of >0.7, which is the value recommended by the developers of the Icity protocol. These Pfams fell into three types: (1) Pfams containing the VR sequence(s), (2) domains fused to RT, and (3) accessory proteins potentially functionally linked to the DGR system. We hypothesize that any of these types may be involved in a piece of DGR-dependent prokaryotic defense machinery.

Prokaryotic immunity systems often co-localize, forming so-called ‘immunity islands’ [18,19,20]. A catalog of the Pfams found in genomic neighborhoods of known immune systems, and therefore potentially involved in prokaryotic antiviral defenses, was recently built [19,20]. Hereafter, we refer to such Pfams as ‘immune Pfams’. We further focused on DGR systems that are functionally coupled to immune Pfams. In total, 8828 out of 32,321 (27%) analyzed DGR systems had in their vicinity an immune Pfam with a high average Icity score. For further analysis, we kept only sequences possessing at least one immune Pfam. We classified all sequences according to Pfam similarity. Figure 2 shows Pfam compositions that occurred more than 20 times in our dataset, where each Pfam had an Icity score of >0.7 in at least one run out of seven. Appendix A contains the complete list of Pfam compositions, where each Pfam had an Icity score of >0.7 in at least one run. Appendix A provides descriptions of Pfams shown in Figure 2.

We should briefly note that Pfam PF05635.13, the most common in the aggregated data, and also occurring in the majority of Pfam compositions, resembles the Avd protein. This protein has no Pfam attribution to date. Nonetheless, the Avd protein from the BPP-1 phage is classified in the CDD database [24] as cd16375, a family that also contains functionally uncharacterized bacterial proteins, some of which are encoded by an atypically large intervening sequence present within some 23S rRNA genes [25]. However, the Pfam database describes the PF05635.13 domain as a 23S rRNA-intervening sequence protein. The sequence and structural alignment for the BPP-1 Avd protein (PDB code 4DWL) and a representative PF05635.13 (PDB code 2GSC) suggest that their homology is found in the NCBI Structure database (https://structure.ncbi.nlm.nih.gov/icn3d/share.html?6MDdtoQPdZu6QeuFA, accessed on 11 March 2023). Other known accessory proteins in DGR systems are HRDC and MSL [12]. The former has been classified as PF00570.25 and is also one of the most common Pfams with a high Icity score (Figure 2). The later, with Pfam attribution PF01624.22, had a high Icity score, although not in all seven runs. These examples demonstrate that our approach is relevant for the biology of DGR systems. To our knowledge, there is only one protein for which the DGR association was suggested [12] that is absent from our analysis because it is not attributed to any Pfam or CDD.

In Figure 2, one can see that several Pfam compositions are subsets of the other ones (with larger Pfam numbers). The larger compositions (supersets) can occur both more frequently and less frequently compared to the occurrence of their subsets found to stay separately. These composition variants may represent different (sub)types of DGR-based immune systems, or our observation may reflect the fact that about 50% of the selected sequences are metagenomic contigs shorter than 20 kbp. The maximum of the distribution (the mode) is 3 kbp (Appendix A). Therefore, Pfam compositions appearing as subsets may be encoded by shorter sequences, lacking the rest of the Pfams from the corresponding superset. Using complete genome sequences would be preferable in order to maximize the chance of observing intact and untruncated DGR-based immune systems. However, the availability of only a small number of such sequences (complete genomes encoding DGR) undermined our effort to find potential immune systems utilizing DGR mutagenesis. Thus, including longer contigs in our analysis presented a reasonable trade-off in this context. To improve the reliability of our predictions regarding the association of Pfams with DGR systems, we further consider only Pfams that had an Icity score of >0.7 in all seven runs. When we visualized these Pfam compositions encoded by contigs longer than 20 kbp, we obtained a qualitatively similar picture, moduloreduced dataset (Appendix A). Supersets of both types (having more common or rare subsets) were preserved, occurring in a relatively uniform manner across the distribution of Pfam compositions. For each subsequent analysis presented here, we considered only the compositions encoded in more than 20 sequences, among which at least four contigs had a length of >20 kbp. Eleven compositions in total satisfied these criteria. Since the majority of these Pfams are poorly characterized, later in this paper we will use the lists of corresponding Pfam accessions to denominate putative DGR systems without considering their possible molecular function in prokaryotic defense.

Functional genetic systems typically demonstrate a more or less conserved order of genes (see [26,27]). A set of *n* Pfams can be organized in n! possible permutations, and each Pfam can be encoded by one of two complementary strands giving $2^{n}$ variants. Since the order of genes in an entire system can arbitrarily be chosen between forward and downstream orders, the total number should be divided by a factor of two. Thus, *n* Pfams would yield $2^{n-1}\cdot n!$ possible arrangements, which translates to four if *n* equals two, and twenty-four if *n* equals three. Referring to Leo Tolstoy: “*All happy families are happy in the same way*”, we will associate functional DGR-based defense systems with happy families and expect similar behavior for different DNA sequences encoding the same Pfam composition. In the case of functional defense systems, one would expect to observe a very limited number of possible variants for the order of genes, where one or maybe two types of organization comprise the majority. Figure 3 shows the proportions of actual Pfam orders related to possible variant numbers for the eleven selected compositions. Only in two Pfam compositions were the most abundant types of organization found in 79% and 90% of the sequences. All other putative DGR-based immune systems demonstrate conserved organization, where >97% of the sequences share the same order and direction of genes. This suggests that the selected Pfam compositions indeed represent functional genetic systems.

Prokaryotic defense systems are often spread via horizontal gene transfer (HGT) [18]. Nevertheless, the events of horizontal gene transfer are relatively rare. The adaptation of a laterally acquired genetic module (e.g., a defense system) to a new host should include, among other factors, the gradual bringing of the codon usage frequencies of the acquired module closer to the new host. Thus, the disparity in the codon usage bias between a putative gene system and the rest of the genome may indicate a recent HGT event. Figure 4 shows the proportion of sequences, for which the Kolmogorov–Smirnov test resulted in a significant *p*-value (<0.05) for eleven selected Pfam compositions. The minimum *p*-value for these compositions was 0.12. Thus, these Pfam compositions demonstrate significant deviations in codon usage from the rest of the sequence by which they are encoded. Although there are quite a few possible reasons for a region of a genome to have a distinctive codon usage pattern, this behavior supports, rather than disproves, the possibility of lateral transfer for these Pfam compositions.

Prokaryotic defense systems also become adopted and repurposed by viruses, e.g., the CRISPR-Cas system [28] and R/M systems [29], which have been found in bacteriophage genomes. However, phages typically possess defense or anti-defense mechanisms distinct from the systems found in bacterial genomes [30]. Although HGT between prokaryotes and their viruses has been documented many times, the functionality and organization of a defense system make it rather suitable either for cells or for viruses. Thus, for each type of putative prokaryotic immune system, one should expect an uneven distribution between cellular and viral genomes. Figure 5 shows the proportion of cellular versus viral (including prophages) sequences encoding putative DGR-based defense systems. The highest fraction of the least abundant localization (cellular or viral) observed among the analyzed Pfam compositions was 25%, which is already significantly biased with respect to the even distribution (50%). The second highest value was 6% (Figure 5). Thus, the selected Pfam compositions can be classified either as predominantly viral or cellular.

The eleven selected putative DGR-based immune systems share only one predicted protein functional category—reverse transcriptase activity. The reverse transcriptases of DGR systems are classified into six major phylogenetic clades [13]. By analogy with the data in the CRISPR-Cas system [26] and other prokaryotic defense machinery [31], one would expect that most of the aforementioned types of putative DGR-based immune systems have a monophyletic origin. Therefore, each type should possess a single type of RT. The presence of a second type of reverse transcriptase within a single type of DGR system is, in theory, possible (due to the lateral transfer events) but less likely in our hypothesis. Additionally, in the case of the polyphyletic origin of a particular type of DGR-based system, accessory proteins would be better suited for one type of reverse transcriptase, thus creating the basis for one type of RT to outcompete the other ones within a given DGR system. Figure 6 demonstrates the proportions of six major RT clades in DGR systems in eleven selected putative defense systems types. Of note, 7 out of 11 systems contain a single RT type (all representatives of a composition belonging to the same DGR clade as defined by [13]). Only for one system the most common RT type does not comprise a majority (occurrence <50%).

The biological significance of the classification of the RT proteins of DGR systems into six major clades may correspond to the taxonomy of the organisms possessing DGR systems to a certain extent. One would expect that, within the same taxon, there should be DGR systems from the same RT clade. Alternatively, a DGR system of one type can be found in different non-related taxa, since a successful event of horizontal gene transfer, once it has occurred, would establish a new lineage of DGR system within a recipient taxon. Thus, we expected to observe a distribution of Pfam compositions that is qualitatively similar to the distribution across major DGR clades. Figure 7 shows the distribution of the selected putative DGR-based defense systems among the major prokaryotic taxa. Figure 8 demonstrates that the major prokaryotic taxa are strongly biased towards one or two major DGR clades, corresponding to the selected Pfam compositions.

Finally, we analyzed the genes’ organization into Pfams compositions that are included within the selected set (see Figure 3 and the inclusion criteria above) and contain more than two Pfams. In total, four compositions satisfied these criteria (Figure 9). The majority of Pfams within these four compositions are poorly characterized or belong to domains with an unknown function. In all four cases, open reading frames are organized in compact operon-like modules.

## 3. Discussion

Roux and colleagues have demonstrated that DGR systems likely shaped the virus–host interactions in multiple taxa and biomes, representing a fundamental mechanism by which viruses and microbes adapt to a permanently changing environment [13]. Furthermore, DGR systems introduce more than 10% of all amino acid substitutions in some organisms [13]. This level of diversity may preemptively adapt defense genes to yet-unknown potential invaders, as in the case of antibodies and T-cell receptors in vertebrates. In this study, we demonstrated that several types of DGR systems co-localize and, therefore, are likely to functionally interact with the known or predicted families of defense proteins [19,20]. These gene systems pass all of the tests we applied for identifying functional modules. The results of our bioinformatic analysis enable the identification of the most promising candidate systems for the experimental evaluation of DGR-associated antiviral activity. The success rate in analogous attempts of new prokaryotic defense systems identification was 10 out of 28 tested candidates and 29 out of 40 as reported in [19,20], respectively.

When initiating this work, we hypothesized a functional mechanism for DGR-mediated immunity, similar to how adaptive immunity works in mammals. In mammals, the immune system recognizes a specific antigen particle using immunoglobulins or T-cell receptors and then removes or deactivates the target. In some of the DGR systems demonstrated in Figure 9, we observed VRs located in predicted Ig-like domains. Ig-like domains are mainly present in exposed proteins in bacteria and phages, and they play a role in interacting with specific targets [32,33,34]. It is noteworthy that some DGR systems have VRs present in exposed proteins that function as bacterial adhesins [14]. However, the phage-specific sequestration of the phage virions by tight binding in a conformation unsuitable for infection may be one of the ways to reduce phage-induced mortality. The periplasmic localization of the immune protein may provide an opportunity to interact with viral proteins involved in the conduction of the DNA to interfere with the viral genome internalization. Interestingly, such mechanisms of the superinfection exclusion were described in phage T4 [35] and in phage HK97 [36], though without any connection to DGRs. Alternatively, binding of the viral proteins by variable cytosolic VR containing pathogen recognition proteins may be coupled with the directing of the target to degradation machinery as it is performed by ATP-dependent protease adaptor proteins [37] or to direct the attack of the viral nucleo-proteid complexes, such as replication initiation complexes, which would lead to the degradation of the phage DNA. However, our data currently provide no hints for a possible mechanism of action of a DGR-based bacterial antiviral system. The experimental identification of the functional DGR-based immune systems among the predicted candidates would allow for elucidation of the underlying mechanism.

## 4. Methods

We have used all of the publicly available sequence data on DGR systems reported in [12,13,21,22]. We downloaded the nucleotide sequences encoding DGR systems along with the annotations from IGM and NCBI [38,39,40]. The ecological and taxonomic data were retrieved from [13].

To detect the conserved protein domains (Pfams), which are potentially functionally linked to the components of DGR systems, we used a modified Icity protocol [23]. This tool implements the ‘guilt by association’ principle, i.e., the proposition that functionally related genes often co-localize in the genomes (for example, genes, involved in a specific metabolic pathway, are frequently found in the same operon). Co-localization of genes in a sequence dataset may happen due to pure chance, depending on the amount of data and the possible overrepresentation of some organisms. These factors require statistical corrections, which are implemented in the Icity algorithm. The Icity protocol performs, as the very first step, the clustering of the protein sequences according to their similarity using a stochastic approach [41]. We replaced this step with clustering according to the Pfam families to which the protein domains belong [42]. The rationale behind the first step’s replacement is to further use data regarding the Pfams associated with known prokaryotic defense machinery (see below). The proteins containing VR were analyzed separately from their non-VR peers from the same Pfam clusters.

All of the downstream analyses were performed according to the original Icity protocol (starting from a psiblast search; see original study) to generate the w${}^{\prime }$/w estimates (see [23]) central to the original protocol. We applied the Icity protocol separately to the DGR components most essential to our opinion (RT and VR). As a result, we obtained protein clusters with their associated Icity scores, a metric reflecting the probability of a functional association between a given protein and the gene of interest (in this study, the components of DGR systems).

Due to the intrinsic stochasticity embedded in the Icity protocol, the results from different runs may not exactly coincide. However, the Icity scores computed for a Pfam in different runs do not deviate significantly. In order to improve the reproducibility of our analysis, we repeated the Icity scoring until no new Pfams with a score of >0.7 (the value recommended by the authors of the Icity protocol [23]) were revealed in two consecutive runs. This criterion was satisfied after seven runs, in which a single run consisted of applying the protocol to RT and VR separately and then taking the maximum value out of two for each Pfam. As a result, for each Pfam, we obtained seven values of Icity scores. We selected the most promising Pfams as the ones with all seven Icity score values of >0.7. Thus, we obtained a number of gene compositions, in which the components of DGR systems functionally interact with other genes with high confidence.

We next focused on gene compositions, in which at least one protein family (domain) was known or predicted to be involved in prokaryotic antiviral defense, according to the recent large-scale studies [19,20]. We visualized the selected gene sets with upsetplot [43]. Proteins containing VR were treated as a separate entity from the other sequences within the same Pfam cluster.

The codon usage bias was determined for all of the open reading frames longer than 360 nucleotides within the putative DGR systems ±10 kbp and for the region of the same size in the most distant part of the same sequence (assuming the sequence is circular). Sequences for which the distance between these two regions was less than 20 kbp were omitted from this analysis. The codon usage bias was estimated with program chips from the EMBOSS package [44,45].

The selected representatives of the highly presented types of gene compositions and those containing more than two Pfams were visualized with clinker [46]. We used R 4.1.2, Python 3.8, and biopython [47] for data handling and visualization.

## Figures and Tables

**Figure 1 ijms-24-05614-f001:**
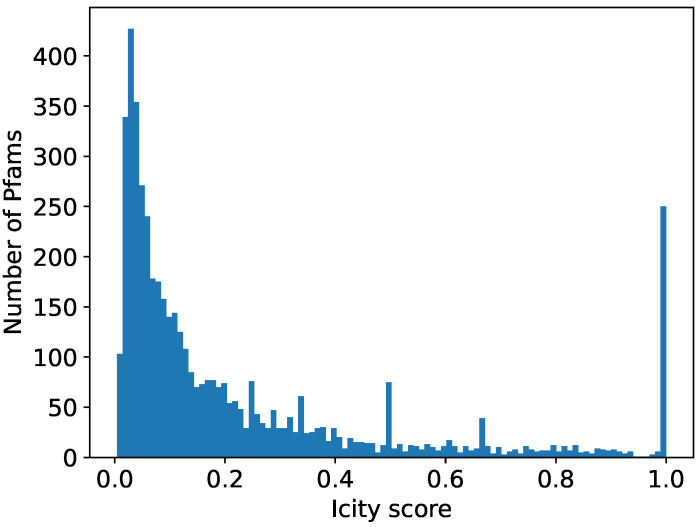
Distribution of Icity scores averaged over seven runs for Pfams found within 10 kbp from components of DGR systems. Icity score shows possibility of functional interaction between given Pfam and DGR systems.

**Figure 2 ijms-24-05614-f002:**
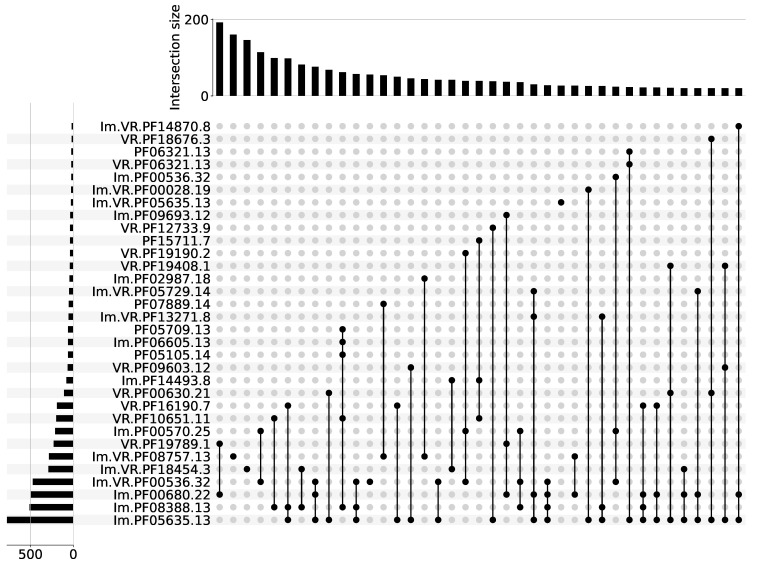
Pfam compositions including at least one immune Pfam. Each Pfam had an Icity score of >0.7 in at least one run out of seven. Pfams associated with known prokaryotic defense systems have the prefix ‘Im.’. Proteins containing variable repeats have the prefix ‘VR.’. The histogram above the diagram shows the occurrence of corresponding Pfam composition in the dataset. The histogram on the left demonstrates the occurrence of the corresponding Pfam in the dataset.

**Figure 3 ijms-24-05614-f003:**
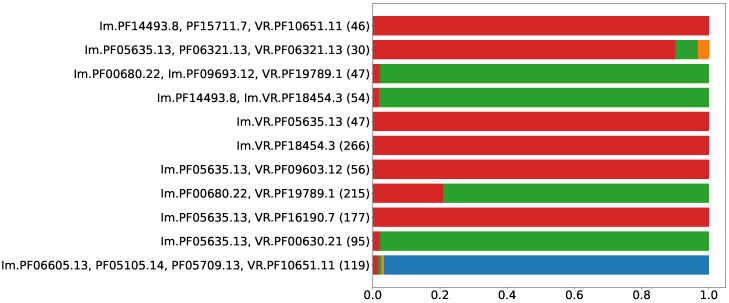
Proportions of gene arrangements among selected Pfam compositions. Numbers in brackets show amount of sequences containing given Pfam composition. Colors show proportions of Pfam arrangements for a composition and are unrelated for different Pfam compositions.

**Figure 4 ijms-24-05614-f004:**
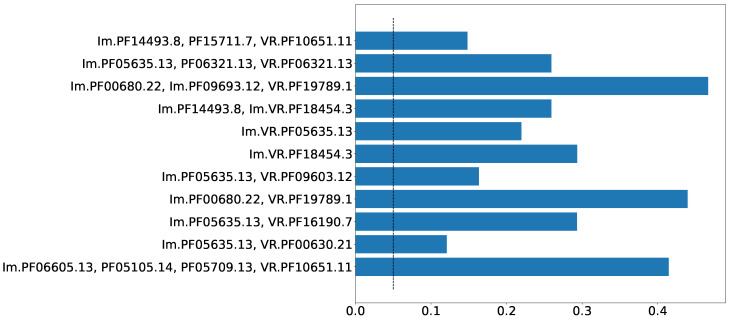
Proportion of sequences, for which codon usage bias in the genes in close neighborhood (±10 kbp) was significantly different (KS *p*-value <0.05) from codon usage bias in the region of the same size, located on the same sequence at the distance of half of the sequence length. This choice for distance guarantees remoteness between the considered regions in the case of circular DNA. Dashed line indicates value of 0.05.

**Figure 5 ijms-24-05614-f005:**
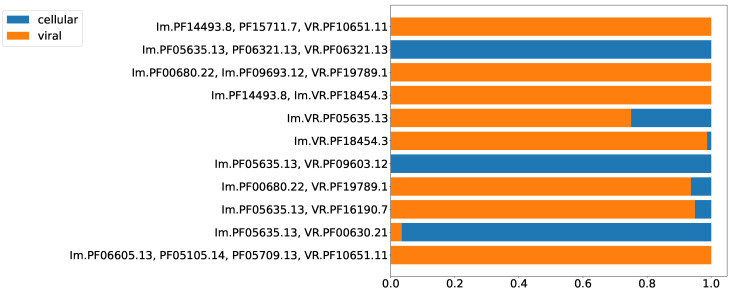
Proportion of viral and cellular residence for selected Pfam compositions.

**Figure 6 ijms-24-05614-f006:**
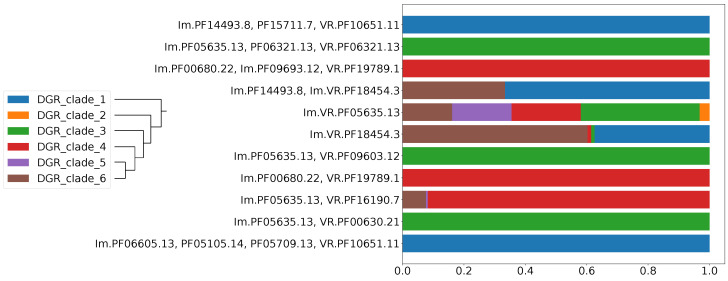
Proportion of major DGR clades based on RT phylogeny in eleven selected Pfam compositions.

**Figure 7 ijms-24-05614-f007:**
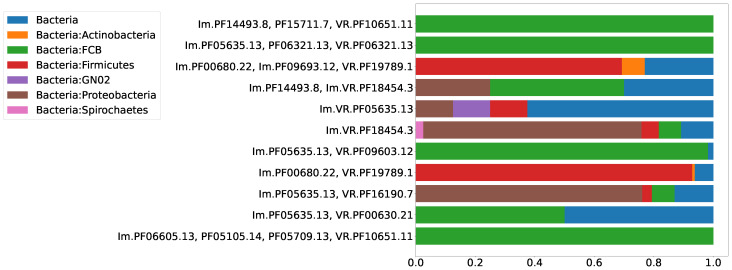
Distribution of eleven selected Pfam compositions among major prokaryotic taxa.

**Figure 8 ijms-24-05614-f008:**
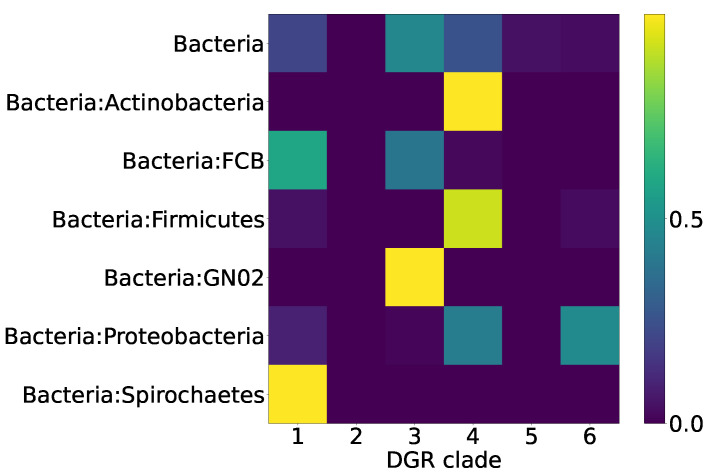
Proportions of major DGR clades within major prokaryotic taxa for eleven selected Pfam compositions.

**Figure 9 ijms-24-05614-f009:**
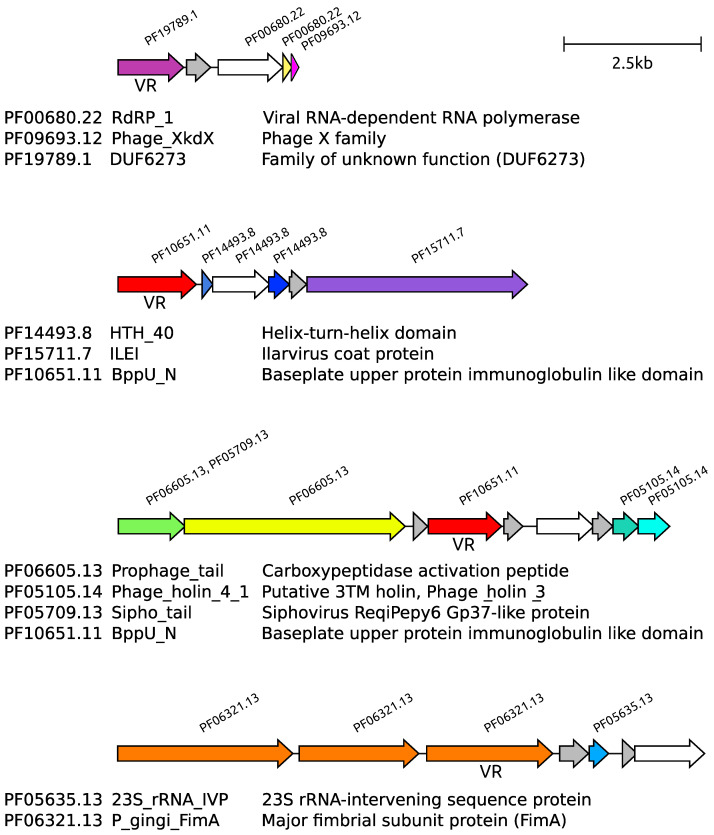
Organization of selected Pfam compositions. Open reading frames are shown as arrows. Reverse transcriptase genes are shown in white. Irrelevant open reading frames are shown in grey. Different genes share the same color if they have more than 30% identical amino acids in blast alignment. Sign VR indicates proteins containing variable repeats.

## Data Availability

Complete instructions, data, and the code to reproduce our results are found in the GitHub repository (https://github.com/ibelalov/DefensiveDGR, accessed 11 March 2023).

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
