# Peer review of "Diversity-Generating Retroelements in Prokaryotic Immunity"

_ijms, 2023, doi:10.3390/ijms24065614_

Round 1

Reviewer 1 Report

Here, Belalov et al. present a study on diversity-generating retroelements (DRGs) and their possible role in adaptive bacterial defenses. The authors first catalog all the proteins containing Pfam in the vicinity of known DGRs and further focus on DGR systems co-localized with known or predicted families of defense proteins. The authors next investigate eleven selected Pfam compositions and show the conservation of gene order of putative DGR-based immune systems. They further demonstrate using codon usage bias analysis that selected Pfam compositions are most likely to move between genomes by lateral transfer and that most sequences encoding putative DGR-based defense systems are of viral origin. The paper's subject perfectly aligns with a current interest in discovering novel bacterial immune strategies. It utilized a successful approach of "guilty by association," which allowed a significant expansion of the arsenal of bacterial defenses in recent years. The approach is legitime and should help decipher adaptive bacterial immunity strategies in the future, for example, by expanding a set of candidate DGR-based defense systems if DGR's neighboring genes would be clustered based on sequence homology, without relying on external domain annotation (Pfam).

Overall the manuscript is well-written and follows a logical flow. However, some results seem preliminary, and some follow-up analysis would improve the study's rigor. The following are specific points to consider:

1. Figure 1: There is a clear high pick around Icity score 1.0, which would be interesting to investigate in more detail; do those Pfams belong to known functions such as, for example, known defense systems, metabolic paths, or receptors?.. Do those Pfams from the pick differ from other Pfams between 0.7 and 1.0 scores?

2. Figure 2: It would be helpful to have additional information on Pfams in Figure 2, such as a short description, for example, Pfam annotation (or HHpred annotation). The authors could add the gene composition of each subset as arrows and mark functional domains (if known) above each arrow (gene). Corresponding descriptions of operons containing Pfams with known functions (especially if they have been previously shown to be involved in immunity) should appear in the text.

3. L127-128: Citing Leo Tolstoy is unnecessary as it does not help to clarify the analysis. Instead, refer to the literature and discuss that most known defense systems consist of two or more genes and that their operon organization is relatively conserved, precisely what the authors see in their analysis.

4. L133: "eleven selected compositions" would be helpful to add a few words about why those eleven were explicitly selected for further analysis. The authors should state the criteria: abundance within microbial genomes, Icity score, or containing domains with known annotation?..

5. Figure 6: For clarity (and visual diversification), it would be helpful to present those results via a phylogenetic tree (with six clades based on RT phylogeny) and eleven selected Pfam compositions represented via arrows in operons with the percentage of occurrence for each clade. This would also allow seeing HGT events more clearly. 

6. L139-151. The authors state that selected Pfam compositions likely are spread via HGT. They use the codon usage bias disparity between the putative gene system and the rest of the genome to measure the lateral transfer of the selected DNA fragments. However, the data supporting the latter mechanisms need to be more rigorous. For example, for a selected set of Pfam compositions, the authors could compare the gene composition of closely related bacterial strains and show that the fragment containing the DGR+defense system is located on the defense island (present/absent in closely related strains). 

7. Figure 9 and discussion under this figure. All four operon-like modules have viral annotation, which is inconsistent with the central hypothesis. Rather than focusing on Pfam compositions containing two or more Pfams, the authors should focus on the genes with homology (or HHpred predictions) to human and plant genes involved in adaptive immunity. Also, a short notice on the different colors of Figure 9 should be provided.

Reviewer 2 Report

The manuscript is generally well written but in many ways feels like observations in search of function. The main hypothesis stated in the abstract "We proposed a hypothesis that prokaryotes employ diversity-generating retroelements to prepare defense proteins against yet-unknown invaders." I did not find to be adequately addressed, rather the authors merely found correlations between locations of VR and other elements.  

I think the manuscript would be significantly strengthened if the authors looked more deeply at some of the these and made at least a suggestive case as to why the diversity elements might be part of a defense mechanism. I'm not even certain from reading the manuscript whether the identified VR elements are even suggestive of being translated.  I recognize that some in the past have been shown to be, but are those the anomaly or the norm.  

Round 2

Reviewer 2 Report

I still think this manuscript would be improved with some changes in the Introduction and Discussion as previously mentioned that provided better context for this work. These were not addressed in the revision. 

Author Response

We have revised the Introduction and Discussion sections to provide a more comprehensive explanation for investigating potential connections between DGR systems and prokaryotic defense, as well as emphasizing the potential involvement of Ig-like domains in this process. These modifications aim to present a compelling argument for the relevance of our study and the significance of our findings.